

# Domain-wall melting and entanglement in free-fermion chains with a band structure

**Viktor Eisler**

Institute of Theoretical and Computational Physics, Graz University of Technology,
Petersgasse 16, A-8010 Graz, Austria
Institute of Physics, University of Graz, Universitätsplatz 5, A-8010 Graz, Austria

viktor.eisler@uni-graz.at

## Abstract

We study the melting of a domain wall in free-fermion chains, where the periodic variation of the hopping amplitudes gives rise to a band structure. It is shown that the entanglement grows logarithmically in time, and the prefactor is proportional to the number of filled bands in the initial state. For a dimerized chain the particle density and current are found to have the same expressions as in the homogeneous case, up to a rescaling of the velocity. The universal contribution to the entropy profile is then doubled, while the non-universal part can be extracted numerically from block-Toeplitz matrices.

# 1 Introduction

The transport properties of quantum many-body systems have been the topic of numerous investigations and continue to receive increased attention [1]. A special focus has been devoted to the dynamics in one-dimensional integrable systems [2], which exhibit various forms of anomalous transport due to their distinguished properties. In particular, the existence of extensive families of long-lived quasiparticle excitations allows to treat their dynamics within a generalized hydrodynamic (GHD) framework [3,4]. The theory of GHD has been a great success in understanding various quantitative features of integrable transport, and its machinery has been developed in various directions, see [5–8] for reviews.

One of the simplest setups to test far-from-equilibrium dynamics in 1D quantum chains is the so-called domain-wall melting [9]. In this protocol one prepares a domain wall, where the orientation of the otherwise homogeneous spins is flipped at a single location, and the system is let evolve freely from this inhomogeneous initial state. In the fermionic analogue of the problem, this corresponds to a step-like initial density, which evolves into a nontrivial profile during melting. The ensuing front region was first described for a free-fermion chain [10], and was later extended to the interacting case using GHD techniques [11]. In general, one finds a ballistic expansion in the Luttinger-liquid regime, except for its boundary, corresponding to the isotropic Heisenberg chain in spin language [12–17].

Besides the studies on the particle density and current, there has been continued interest in the characterization of particle-number fluctuations and entanglement along the front. In fact, both of them were found to grow logarithmically in time for free-fermion chains [18–20], and the emerging profiles were later understood using techniques of conformal field theory (CFT) [21,22]. In essence, the description can be understood as a quantized version of the GHD solution, and the emerging state is effectively captured by an inhomogeneous Luttinger liquid [23,24]. The quantum GHD treatment can also be applied to derive entanglement profiles for more complicated initial states, such as the double domain wall [25–27]. Further results on domain-wall melting have also been obtained via hydrodynamic arguments in free-fermion related spin chains [28–34].

While the hydrodynamic approach provides an excellent description for initial states with inhomogeneities, there is much less known about the case where the time evolution operator itself breaks translation invariance. The simplest example is a single localized defect in an otherwise homogeneous chain. This leads to the scattering of incoming wave modes, and the resulting density and current profiles can be worked out in the noninteracting case [35–37]. Remarkably, for a domain-wall initial state, the entropy growth becomes linear due to the entanglement between transmitted and reflected wave components [38–44], leading to extensive long-range entanglement in the steady state [45,46].

Here we study free-fermion chains with a different type of inhomogeneity, with the hopping amplitudes given by a periodic pattern. The time-evolution operator has thus a $p$-site translation invariance, which leads to a band structure in the dispersion, with the bands typically separated by a gap. Starting from a domain wall, we show that the entanglement growth remains logarithmic, however with a prefactor that is multiplied by the number $p$ of filled quasiparticle bands. Moreover, focusing on the case $p = 2$ which corresponds to a dimerized chain, the hydrodynamic limit of the density and current profiles are simply obtained from the $p = 1$ case by rescaling time with the maximum quasiparticle velocity. Although this relation also yields the universal part of the entanglement profile, the contribution depending on the lattice structure of correlations must be extracted numerically, with analytical results available only in the fully dimerized limit of the chain.

The rest of the manuscript is structured as follows. In Section 2 we introduce the setup for the domain-wall melting in a dimerized chain. The time evolution of the correlations in the hydrodynamic limit is presented in Sec. 3, with a focus on the particle density and current. This is followed by the study of entanglement spreading in Sec. 4, addressing the entropy profile as well as temporal oscillations. Finally, some results for a three-band model are presented in Sec. 5. The manuscript concludes with a discussion in Sec. 6, while various details of the calculations are reported in three appendices.

## 2 Model and setup

The dimerized hopping chain is described by the Hamiltonian

$$\hat{H} = - \sum_{m=-N/2+1}^{N/2} \left( \frac{1+\delta}{2} c^\dagger_{2m-1} c_{2m} + \frac{1-\delta}{2} c^\dagger_{2m} c_{2m+1} + \text{h.c.} \right), \tag{1}$$

where $c_j$ and $c^\dagger_j$ are fermionic creation and annihilation operators on a periodic chain with $2N$ sites. The model can be diagonalized by first introducing fermion operators on the two sublattices, $c_{2m-1} = a_m$, $c_{2m} = b_m$, and their corresponding Fourier modes

$$\begin{pmatrix} a_m \\ b_m \end{pmatrix} = \frac{1}{\sqrt{N}} \sum_k e^{ikm} \begin{pmatrix} a_k \\ b_k \end{pmatrix}, \tag{2}$$

where $k = \frac{2\pi}{N} l$ is the sublattice momentum with $l = -N/2+1, \ldots, N/2$. In the next step, one introduces new operators via

$$a_k = \frac{1}{\sqrt{2}} e^{-i\varphi_k/2} (\alpha_k + \beta_k), \qquad b_k = \frac{1}{\sqrt{2}} e^{i\varphi_k/2} (\alpha_k - \beta_k), \tag{3}$$

which bring the Hamiltonian into the diagonal form

$$\hat{H} = - \sum_k \omega_k \left( \alpha^\dagger_k \alpha_k - \beta^\dagger_k \beta_k \right), \tag{4}$$

where the phase and the dispersion are given by

$$e^{i\varphi_k} = e^{i\frac{k}{2}} \frac{\cos \frac{k}{2} - i\delta \sin \frac{k}{2}}{\omega_k}, \qquad \omega_k = \sqrt{\cos^2 \frac{k}{2} + \delta^2 \sin^2 \frac{k}{2}}. \tag{5}$$

One thus arrives at a two-band free-fermion Hamiltonian with a band gap $2|\delta|$. Since the phase and dispersion in (5) depend only on the halved momentum $q = k/2$, it will be useful to work with this variable, defined on the reduced Brillouin zone $q \in [-\pi/2, \pi/2]$.

We will be interested in the dynamics generated by the quadratic Hamiltonian (1). Assuming a Gaussian initial state $|\psi(0)\rangle$, all the information about the time-evolved state is encoded in the two-point correlation matrix $C(t)$, with matrix elements given by

$$C_{m,n}(t) = \langle \psi(t) | c^\dagger_m c_n | \psi(t) \rangle, \qquad |\psi(t)\rangle = e^{-i\hat{H}t} |\psi(0)\rangle. \tag{6}$$

In the following we will focus exclusively on the domain-wall initial state, where the sites on the left/right hand side of the chain are completely filled/empty. The time evolution of the correlation matrix is then given by

$$C(t) = U^\dagger C(0) U, \qquad C(0) = \begin{pmatrix} \mathbb{1} & 0 \\ 0 & 0 \end{pmatrix}. \tag{7}$$

Here $\mathbb{1}$ denotes the $N \times N$ identity matrix, and the propagator $U = \mathrm{e}^{-iHt}$ is obtained from the $2N \times 2N$ hopping matrix $H$ corresponding to the Hamiltonian (1) in the local site basis. Note that in our analytical calculations we shall consider the thermodynamic limit $N \to \infty$, obtained from the above formulation assuming periodic boundary conditions. In contrast, in our numerics we consider an open chain, to avoid the doubling of the front region in the melting of the domain wall.

Before starting our analysis, we first introduce the main physical quantities of interest. In particular, we shall evaluate the fermion density and current, obtained respectively as

$$\rho(n,t) = \langle \psi(t) | c_n^\dagger c_n | \psi(t) \rangle , \qquad J(n,t) = 2t_n \, \mathrm{Im} \, \langle \psi(t) | c_n^\dagger c_{n+1} | \psi(t) \rangle , \qquad (8)$$

where $t_{2n-1} = (1+\delta)/2$ and $t_{2n} = (1-\delta)/2$ are the dimerized hopping amplitudes. Note that these are immediately related to the (diagonal and nearest-neighbour) correlation matrix elements in (6). Furthermore, we are also interested in the spreading of the entanglement entropy between a subsystem $A = [-N+1, n_0]$ and its remainder, which follows as

$$S(n_0, t) = -\mathrm{Tr}\left[ C_A(t) \ln C_A(t) + (1 - C_A(t)) \ln(1 - C_A(t)) \right], \qquad (9)$$

where $C_A(t)$ is the reduced correlation matrix, with elements (6) restricted to $m, n \in A$.

## 3 Correlations and hydrodynamics

In the following we analyze the time evolution of the correlations (6) in the hydrodynamic regime, obtained by letting $m, n, t \to \infty$ with their ratios $m/t$ and $n/t$ kept fixed. Due to the two-site shift invariance of the time-evolution operator, it is useful to introduce a block-matrix notation

$$\mathbf{C}_{m,n}(t) = \begin{pmatrix} C_{2m-1,2n-1}(t) & C_{2m-1,2n}(t) \\ C_{2m,2n-1}(t) & C_{2m,2n}(t) \end{pmatrix}, \qquad (10)$$

where the entries belonging to cell $m$ and $n$ are grouped together. Adopting the same notation for the propagator $U = \mathrm{e}^{-iHt}$, its matrix elements can be found using the transformations in Sec. 2 that diagonalize $H$, which leads to

$$\mathbf{U}_{m,n} = \int_{-\pi/2}^{\pi/2} \frac{dq}{\pi} \mathrm{e}^{iq(2n-2m)} \begin{pmatrix} \cos(\omega_q t) & i \sin(\omega_q t) \mathrm{e}^{i\varphi_q} \\ i \sin(\omega_q t) \mathrm{e}^{-i\varphi_q} & \cos(\omega_q t) \end{pmatrix}. \qquad (11)$$

Carrying out the matrix multiplications in (7) and subsequently taking the hydrodynamic limit is a relatively straightforward exercise, with the details presented in Appendix A. In turn, one arrives at

$$\mathbf{C}_{m,n}(t) = \int_{q_-}^{q_+} \frac{dq}{2\pi} \mathrm{e}^{iq(2n-2m)} \begin{pmatrix} 1 & \mathrm{e}^{i\varphi_q} \\ \mathrm{e}^{-i\varphi_q} & 1 \end{pmatrix} + \int_{-q_+}^{-q_-} \frac{dq}{2\pi} \mathrm{e}^{iq(2n-2m)} \begin{pmatrix} 1 & -\mathrm{e}^{i\varphi_q} \\ -\mathrm{e}^{-i\varphi_q} & 1 \end{pmatrix}, \quad (12)$$

where $q_\pm(m,n,t)$ are the solutions of $v_{q_\pm} = (m+n-1)/t$, with the single-particle group velocities defined as

$$v_q = -\frac{d\omega_q}{dq} = (1-\delta^2)\frac{\sin q \cos q}{\omega_q} . \qquad (13)$$

The result (12) has a very clear interpretation in terms of the quasiparticle picture. Indeed, the two terms simply correspond to contributions from the lower and upper band of quasiparticles $\alpha_q$ and $\beta_q$, respectively. Due to the symmetry of the dispersion in (4), these modes propagate ballistically with velocities $\pm v_q$, and contribute to the correlations if they

reach the midpoint between the cells $m$ and $n$. Note also that (12) can be rewritten more compactly as

$$\mathbf{C}_{m,n}(t) = \int_{q_-}^{q_+} \frac{dq}{\pi} \begin{pmatrix} \cos q(2n-2m) & i\sin\left[q(2n-2m)+\varphi_q\right] \\ i\sin\left[q(2n-2m)-\varphi_q\right] & \cos q(2n-2m) \end{pmatrix},$$ (14)

and it is easy to see that correlations between sites with an even/odd distance are always purely real/imaginary.

## 3.1 Density and current

We first evaluate the particle density and current defined in (8). Note that the result is going to depend only on $|\delta|$, and we assume $0 < \delta < 1$ to simplify notation. The maximal quasiparticle velocity can be obtained via the solution of $\frac{dv_q}{dq} = 0$, which yields

$$\cos^4 q_* - \delta^2 \sin^4 q_* = 0 \quad \to \quad \cos^2 q_* = \frac{\delta}{1+\delta},$$ (15)

and inserting this into (13), the maximum is given by

$$v_{max} = v_{q_*} = 1 - \delta.$$ (16)

This immediately yields the half-size of the front as $(1-\delta)t$. To obtain the density at site $n$, one needs to find the time evolution of the Fermi points $q_\pm(n,t)$, given by the solution of $v_{q_\pm} t = n$. Introducing $v = n/t$, one has to solve

$$\frac{(1-\delta^2)^2}{4}\sin^2(2q_\pm) = v^2\left[\frac{1+\delta^2}{2} + \frac{1-\delta^2}{2}\cos(2q_\pm)\right],$$ (17)

which has roots

$$\cos 2q_\pm = \frac{-v^2 \mp \sqrt{v^4 - 2(1+\delta^2)v^2 + (1-\delta^2)^2}}{1-\delta^2}.$$ (18)

Using trigonometric identities, this can be rewritten as

$$\cos(q_+ - q_-)\cos(q_+ + q_-) = -\frac{v^2}{1-\delta^2},$$ (19)

$$\sin(q_+ - q_-)\sin(q_+ + q_-) = \sqrt{\left(1 - \frac{v^2}{(1-\delta)^2}\right)\left(1 - \frac{v^2}{(1+\delta)^2}\right)},$$ (20)

which has solutions

$$q_+ - q_- = \arccos\left(\frac{v}{1-\delta}\right), \qquad q_+ + q_- = \pi - \arccos\left(\frac{v}{1+\delta}\right).$$ (21)

The Fermi points $q_\pm$ are shown in Fig. 1. Note that the two branches join at $q_*$ obtained from (15), and in the limit $\delta \to 0$ the larger Fermi point remains constant $q_+ \to \pi/2$.

With the expression (21) at hand, the density follows immediately as

$$\rho(n,t) = \frac{q_+ - q_-}{\pi} = \frac{1}{\pi}\arccos\left(\frac{n}{v_{max}t}\right),$$ (22)

which is nothing but the result for the domain-wall melting in a homogeneous chain [10], up to a rescaling by the front velocity $v_{max}$. The expression of the current in (8) depends on the

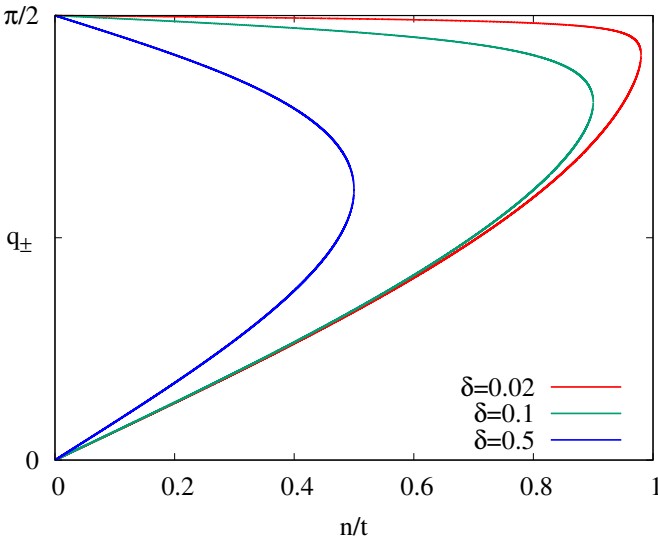

Figure 1: Fermi points $q_\pm$ as a function of $n/t$, for various dimerizations $\delta$.

parity of $n$, and the respective matrix element is contained in $\mathbf{C}_{n,n}(t)$ or $\mathbf{C}_{n,n+1}(t)$ for odd or even $n$. The corresponding off-diagonal entries in the integrand of (14) read

$$\sin\varphi_q = (1-\delta)\frac{\sin q \cos q}{\omega_q} = \frac{v_q}{1+\delta}\,, \qquad \sin(2q-\varphi_q) = (1+\delta)\frac{\sin q \cos q}{\omega_q} = \frac{v_q}{1-\delta}\,. \qquad (23)$$

These factors have to be multiplied by the respective hopping amplitudes $2t_n = 1 \pm \delta$, and one thus obtains for the current

$$J(n,t) = \int_{q_-}^{q_+} \frac{\mathrm{d}q}{\pi}\, v_q = \frac{v_{max}}{\pi}\sqrt{1-\left(\frac{n}{v_{max}t}\right)^2}\,. \qquad (24)$$

Hence, despite the lack of translational invariance of the time-evolution operator, both the density and the current becomes a smooth function of the position, without any parity dependence. In fact, it is easy to check that the continuity equation is satisfied

$$\partial_t \rho(n,t) + \partial_n J(n,t) = 0\,. \qquad (25)$$

The hydrodynamic limit results (22) and (24) are compared to numerical calculations on a finite-size open chain with $N = 300$ in Fig. 2, finding an excellent agreement.

## 4 Entanglement spreading

We now turn our attention to the evolution of the entanglement entropy. As we have seen in the previous section, the results for the particle density and current are intimately related to the domain-wall melting problem in a homogeneous chain. Before presenting the corresponding relation for the entropy, we shall briefly recap the derivation of the result in the homogeneous ($\delta = 0$) case, using the machinery of CFT.

### 4.1 CFT results

In order to understand the spreading of entropy during domain-wall melting, one has to first identify the underlying inhomogeneous field theory. This has been accomplished in [21], by

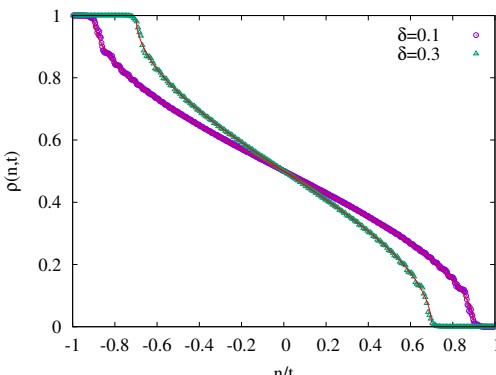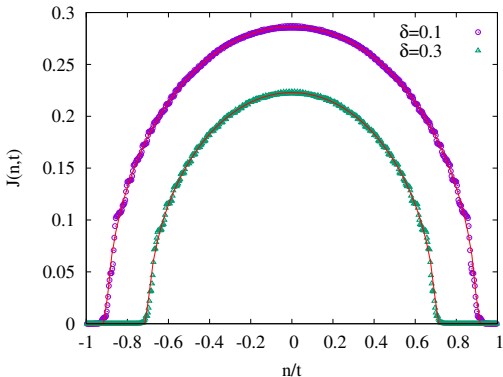

Figure 2: Particle density (left) and current (right) profiles at $t = 300$ for various dimerizations. The symbols are the numerical data obtained for an open chain with $N = 300$, while the red solid lines show the hydrodynamic limits (22) and (24), respectively.

considering the imaginary-time version of the quench problem. Setting $y = it$, the field theory is then defined on an infinite strip with $y \in [-R, R]$, and the initial state becomes a boundary condition along the edges of the strip $y = \pm R$. The field then fluctuates only within a domain given by $x^2 + y^2 < R$, and is completely frozen outside of this disk, which is the famous arctic circle phenomenon [47]. The proper field theory describing this setup has been identified as a Dirac theory in curved space, with the underlying metric given by [21]

$$\mathrm{d}s^2 = \mathrm{d}x^2 + \frac{2xy}{R^2 - y^2}\mathrm{d}x\mathrm{d}y + \frac{R^2 - x^2}{R^2 - y^2}\mathrm{d}y^2. \tag{26}$$

Moreover, it has been shown that this metric is Weyl-equivalent to a flat one, $\mathrm{d}s^2 = \mathrm{e}^{2\sigma}\mathrm{d}z\mathrm{d}\bar{z}$, where the isothermal coordinates and the Weyl factor are given by [21]

$$z(x, y) = \arccos\frac{x}{\sqrt{R^2 - y^2}} - i\,\mathrm{artanh}\,\frac{y}{R}, \qquad \mathrm{e}^{\sigma} = \sqrt{R^2 - x^2 - y^2}. \tag{27}$$

In turn, the effective Euclidean action describing our setup is that of a massless Dirac field with the appropriate background metric

$$\mathcal{S} = \frac{1}{2\pi}\int \mathrm{d}z\mathrm{d}\bar{z}\,\mathrm{e}^{\sigma(z,\bar{z})}\left[\psi_R^\dagger\overleftrightarrow{\partial}_{\bar{z}}\psi_R + \psi_L^\dagger\overleftrightarrow{\partial}_z\psi_L\right]. \tag{28}$$

Having identified the inhomogeneous CFT, the next step is to apply the replica trick and twist-field machinery [48, 49] for the computation of the entropy of the domain $A = [-\infty, x]$, which has been carried out in [22]. The key is to map the inhomogeneous CFT (28), defined on a strip $[0, \pi] \times \mathbb{R}$ in terms of the isothermal coordinates (27), into a flat one living on the upper half plane. This can be achieved by the subsequent application of a Weyl-transformation $\mathrm{e}^{\sigma(z,\bar{z})}$, and the exponential map $g(z) = \mathrm{e}^{iz}$. The twist field inserted at $(x, y)$ in the original geometry is thus mapped into $g(z)$ on the flat upper half plane, which amounts to calculating the entropy of a segment of length $\mathrm{Im}\,g(z)$ located at the boundary of a half-infinite chain. Taking into account the Jacobians of the conformal transformations, one obtains an effective length scale and the universal contribution to the entropy is given by [22]

$$S_u = \frac{1}{6}\ln\left[\mathrm{e}^{\sigma(z,\bar{z})}\left|\frac{\mathrm{d}g(z)}{\mathrm{d}z}\right|^{-1}2\,\mathrm{Im}\,g(z)\right]. \tag{29}$$

Note that we have explicitly included a factor two, which is due to the method of images when calculating expectation values on the upper half plane.

Up to this point we have been working with the Euclidean formulation of the problem, where $y$ is assumed to be real and $R$ has been introduced as an imaginary-time regulator. In order to obtain the result in real time, one first inserts the expressions into the argument of the logarithm in (29), using (27) and the form of the mapping $g(z)$. In a next step, one can analytically continue the result by setting $y = it$ and sending the regulator to $R \to 0$. This yields

$$S_u = \frac{1}{6} \ln \left[ 2 \frac{R^2 - x^2 - y^2}{\sqrt{R^2 - y^2}} \right] = \frac{1}{6} \ln[2t(1 - v^2)], \tag{30}$$

where we defined $v = x/t$.

However, this is not the end of the story, as we have to deal with a lattice problem, which introduces a space-dependent non-universal contribution to the entropy. Nevertheless, this can be found by applying a local density approximation and using the known results for the constant term in a homogeneous chain [50]. Indeed, the time-evolved state is locally described by a boosted Fermi sea $[k_-, k_+]$ with boundaries $k_\pm = \frac{\pi}{2} \pm \arccos v$, such that the lattice contribution is given by

$$S_0 = \frac{1}{6} \ln \left[ 2 \sin \frac{k_+ - k_-}{2} \right] + \frac{s_0}{2}, \tag{31}$$

where $s_0 \approx 0.495$ is the constant calculated in [50]. Note that there is an overall factor $1/2$ due to the effective semi-infinite geometry that follows from the CFT treatment. Inserting explicitly $k_\pm$ and adding the two contributions, one finally arrives at

$$S = S_u + S_0 = \frac{1}{6} \ln \left[ t(1 - v^2)^{3/2} \right] + c_0, \tag{32}$$

with $c_0 = s_0/2 + \ln(2)/3$.

## 4.2 Entropy profile

We are now ready to generalize the treatment presented above to the dimerized chain. The essential input we need, established in Sec. 3, is that the hydrodynamic description of the density and current is identical to the $\delta = 0$ case, up to a rescaling of the maximal velocity. However, to understand the behaviour of the entropy, one should keep in mind that this structure emerges from the contributions of two completely filled bands of quasiparticles. Furthermore, since the initial state has no correlations, one naturally expects those two bands to contribute independently to the entropy. Naively, for $t \gg \delta^{-1}$ this should lead to a doubling of the entropy with respect to the $\delta = 0$ case, once the rescaling of the time $t \to v_{max} t$ has been taken into account. The entropy profile at time $t = 200$ for subsystems $A = [-N + 1, n_0]$ and various $\delta$ is shown in Fig. 3, plotted against the scaling variable $\zeta = n_0/(v_{max} t)$. Although the doubling effect due to the two bands is clearly visible, there is some nontrivial dimerization dependence of the profile, in particular for small values of $\delta$, which is not absorbed by $\zeta$.

The origin of this deviation can be understood by a closer inspection of the correlation matrix in the hydrodynamic limit. Indeed, assuming $q_\pm(\zeta)$ to be fixed, in the spirit of LDA, by the value of the Fermi points around the entanglement cut, (12) has a block-Toeplitz structure with Fisher-Hartwig singularities. Considering a segment of size $L$ of such a block-Toeplitz matrix, the corresponding entropy is expected to have the form

$$S(L, \zeta) = \frac{2}{3} \ln L + 2S_0(\zeta). \tag{33}$$

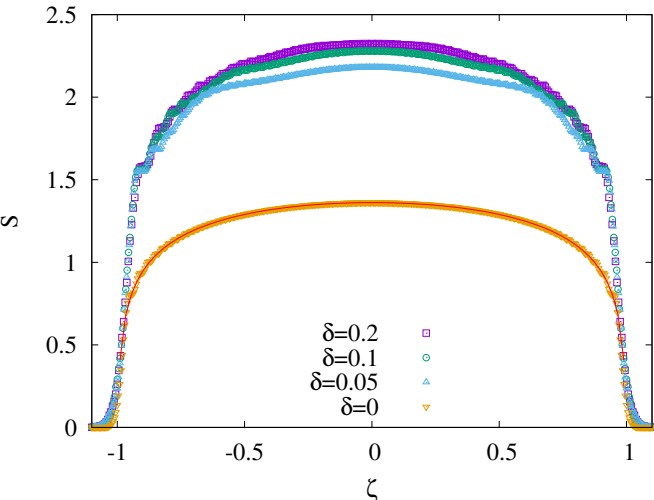

Figure 3: Entropy profile at $t = 200$ as a function of the scaling variable $\zeta$, for various dimerizations $\delta$. The red line shows the analytical result (32) for the $\delta = 0$ case.

Here the coefficient of the logarithm is due to the two disconnected Fermi seas $[q_-, q_+]$ and $[-q_+, -q_-]$ in terms of the reduced momentum, which is known to give a doubling for simple Toeplitz matrices [51, 52]. Note that, even though $q_-(\zeta) \to 0$ for $\zeta \to 0$ such that the two Fermi seas merge, the jump singularities still persist due to sign change in the offdiagonal elements in (12). As shown in Appendix B, the leading logarithmic term in (33) can be verified using recent results on block-Toeplitz determinants with Fisher-Hartwig singularities [53–55]. However, the non-universal piece $S_0(\zeta)$ in the entropy profile is given by the halved subleading constant, for which one has no explicit result available, and has to be extracted numerically.

We thus expect that the entropy profile is given by $S = S_u + S_0(\zeta)$, where the universal contribution is twice the result (30) for $\delta = 0$, after rescaling the time with $v_{max}$, which gives

$$S_u = \frac{1}{3} \log\left[2 v_{max} t (1 - \zeta^2)\right]. \tag{34}$$

In order to check our ansatz, we subtract the universal contribution and compare $S - S_u$ to the function $S_0(\zeta)$ obtained numerically from the entropy (33) of block-Toeplitz correlation matrices. As shown in Fig. 4, the agreement between the data sets is very good, up to oscillations with a characteristic frequency that becomes slower as $\delta \to 0$. Moreover, in the opposite limit $\delta \to 1$, it is possible to find the analytical form of the entropy function

$$\lim_{\delta \to 1} S_0(\zeta) = \frac{1}{3} \ln \sqrt{1 - \zeta^2} + s_0. \tag{35}$$

This result can be obtained indirectly, by evaluating the particle number fluctuations [38–40, 56], which is a simple quadratic function of the correlation matrix $C_A$. As shown in Appendix C, the fluctuations for $\delta \to 1$ are twice the value for a homogeneous ground state in a segment $L/2$ and at filling $q_F = q_+ - q_-$. Applying this relation to the entropy yields (35), which is shown by the red dashed line in Fig. 4, and gives a very good approximation already for $\delta = 0.5$.

## 4.3 Oscillations

Finally, we have a closer look at the oscillations observed in the entropy profile. For simplicity, we consider the bipartition in the middle of the chain, $\zeta = 0$, and have a look at the oscillations

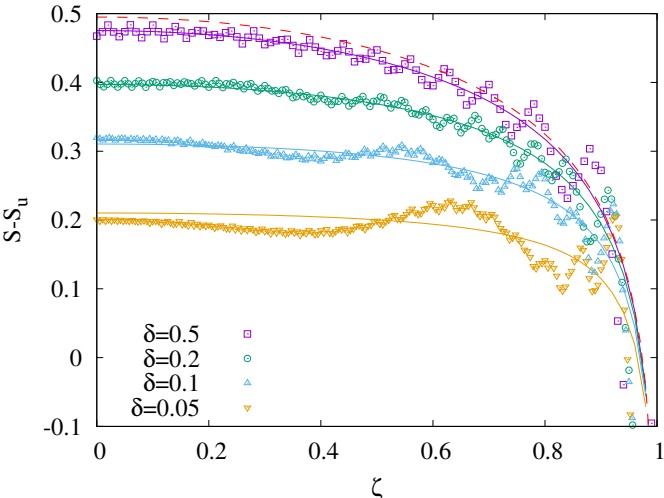

Figure 4: Entropy profile with the universal contribution (34) subtracted (symbols), and compared to the function $S_0(\zeta)$ in (33) obtained numerically (solid lines). The red dashed line shows the result (35) obtained in the limit $\delta \to 1$.

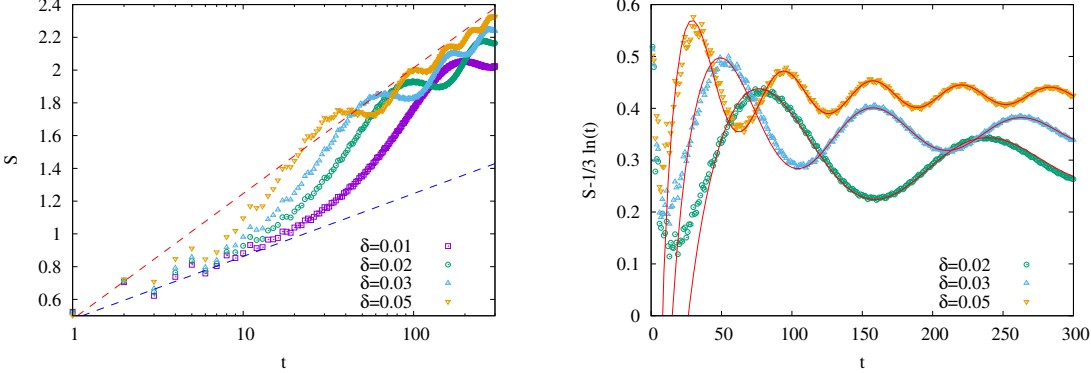

Figure 5: Left: entanglement entropy for a half-chain as a function of time and various $\delta$. The blue dashed line shows the function (32) with $\nu = 0$, while the red one has a doubled prefactor 1/3. Right: oscillatory part of the entropy, fitted with the ansatz (36) as shown by the red lines.

as a function of time for $\delta \ll 1$. This is shown on the left of Fig. 5 on a logarithmic time scale, where the blue dashed line shows the $\delta = 0$ result in (32). Clearly, for $t \ll \delta^{-1}$ the data follows the homogeneous-chain result, i.e. the presence of the gap is not yet visible on these time scales. For larger times one observes a crossover to the asymptotic result with a doubled slope 1/3, indicated by the red dashed line. The superimposed oscillations are visualized on the right of Fig. 5, by subtracting the dominant $1/3 \ln t$ term. They seem to be very well described by the ansatz

$$S - \frac{1}{3} \ln t \approx \alpha + \beta \frac{\cos(2\delta t + \phi)}{\delta t}, \tag{36}$$

which is shown by the red solid lines after fitting the parameters $\alpha, \beta$ and $\phi$. One finds a nice agreement, which shows that the frequency of the decaying oscillations is given exactly by the size of the band gap.

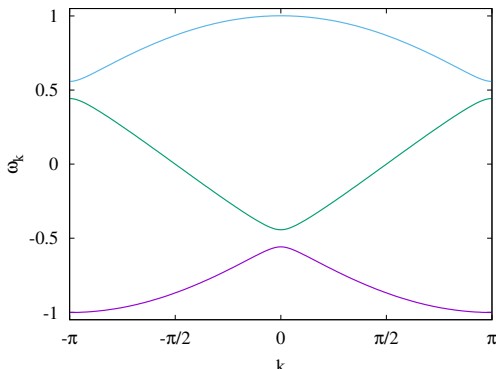 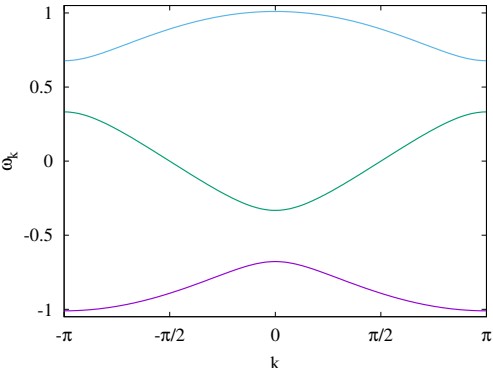

Figure 6: Dispersion of the trimerized chain with hopping amplitudes (38) and $\delta = 0.1$ (left) as well as $\delta = 0.3$ (right), plotted against the sublattice momentum.

## 5 Three-band model

To illustrate the generic behaviour for free-fermion models with multiple bands, we now study a hopping chain with a unit cell of three sites, given by the Hamiltonian

$$\hat{H} = -\sum_{m=1}^{3N} t_m \left( c_{m+1}^\dagger c_m + c_m^\dagger c_{m+1} \right). \tag{37}$$

For simplicity, we choose hopping amplitudes with only one free parameter as

$$t_{3m-2} = (1+\delta)/2, \qquad t_{3m-1} = 1/2, \qquad t_{3m} = (1-\delta)/2. \tag{38}$$

Similarly to the dimerized case, the Hamiltonian can be diagonalized by a Fourier transform and a subsequent unitary that mixes the sublattice degrees of freedom. In turn, this yields a Hamiltonian with three bands

$$\hat{H} = \sum_{\sigma=1}^{3} \sum_k \omega_{\sigma,k} \alpha_{\sigma,k}^\dagger \alpha_{\sigma,k}, \tag{39}$$

where $\alpha_{\sigma,k}$ are the quasiparticle modes with corresponding dispersions $\omega_{\sigma,k}$. The band structure for two different values of the parameter $\delta$ is shown in Fig. 6, with gaps opening at sublattice momentum $k = 0$ and $k = \pm\pi$, and increasing with $\delta$.

Instead of a detailed analysis as in the dimerized case, we now only focus on the main features of the entropy spreading during domain-wall melting. It is easy to guess that, to leading order, the half-chain entropy should now be given by

$$S = \frac{1}{2}\ln t + \text{const.}, \tag{40}$$

which is just three times the $\delta = 0$ result (32), reflecting the contributions from three completely filled bands. As shown in Fig. 7, the numerical results agree very well with the ansatz (40), up to oscillations decaying with time.

Finally, let us have a look at the entropy profile, shown in the left of Fig. 8, together with the corresponding density profile on the right. The entropy has a a jump in the interior of the profile, which can be understood from the structure of the dispersion in Fig 6. Indeed, the three bands are now not identical, and the middle one delivers the global maximum $v_1$ of the velocity, while the upper/lower bands deliver a local maximum $v_2 < v_1$. Hence, the outermost

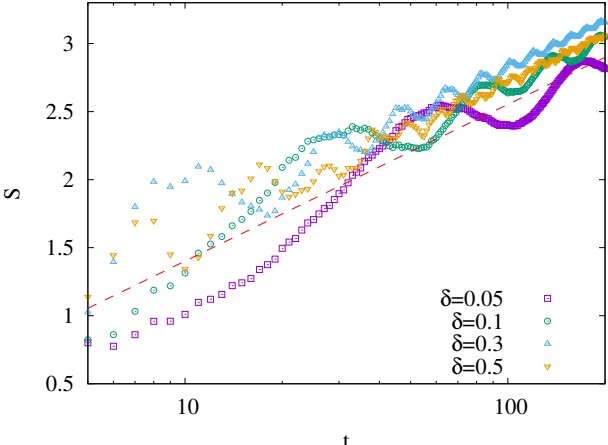

Figure 7: Evolution of the half-chain entropy in the trimerized chain, plotted on a logarithmic time scale. The red dashed line has slope 1/2, corresponding to the result (40).

part $v_2 < |v| < v_1$ of the profile corresponds to contributions from the middle band only. The same feature can also be observed in the density, which develops a superimposed edge regime around $|v| = v_2$, where the fastest quasiparticles from the upper/lower bands start to contribute. Note also that the density develops a fine structure, with different hydrodynamical limits on the sublattices. This indicates that the analogue of eq. (12) must possess a more complicated structure, with non-uniform entries along the diagonals of the corresponding $3 \times 3$ matrices in the integrals.

Comparing entropy profiles at different times, one can verify that it is described by

$$S = \begin{cases} \frac{1}{2} \ln \left[ t \, f(v) \right], & |v| < v_2, \\ \frac{1}{6} \ln \left[ t \, g(v) \right], & v_2 < |v| < v_1, \end{cases} \tag{41}$$

with some unknown scaling functions $f(v)$ and $g(v)$. For more general hopping amplitudes without any symmetry, we expect that each band has its own maximum velocity, and the profile shows three distinct scaling regimes with coefficients being multiples of 1/6.

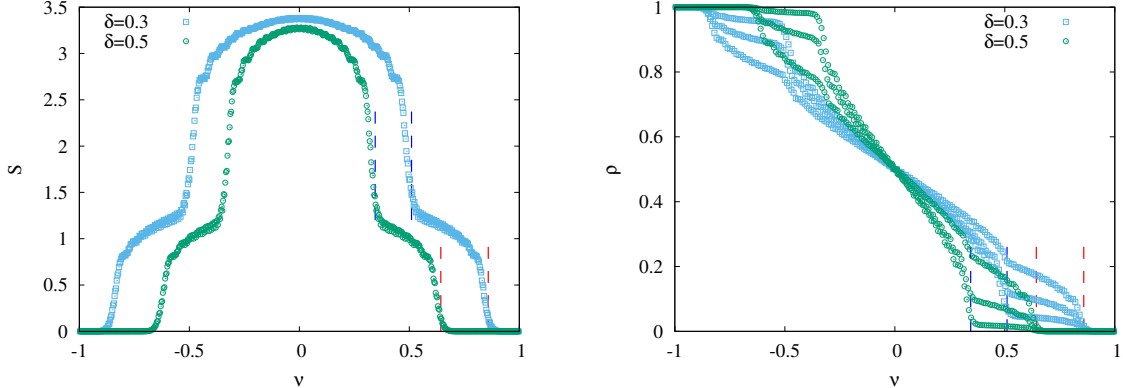

Figure 8: Entropy (left) and density (right) profiles at time $t = 300$ in a trimerized chain with $N = 200$ cells, as a function of $v = n_0/t$ and for different values of $\delta$. The dashed lines indicate the maxima of the quasiparticle velocities $v_1$ in the middle (red) and $v_2$ in the upper/lower bands (blue).

# 6 Discussion

We have studied the melting of a domain wall in free-fermion chains with periodic hopping amplitudes, which give rise to a band structure of the time-evolution operator. The initial state then corresponds to all the quasiparticle bands being filled/empty on the left/right hand side of the chain, and the evolution of the local occupations is governed by hydrodynamics. The half-chain entanglement grows logarithmically, each of the bands contributing a factor $1/6 \ln t$. In particular, for dimerized hoppings we find that the densities and currents are obtained by a simple rescaling of the velocity with respect to the homogeneous case, which allows us to find the universal contribution to the entanglement profile analytically. However, the non-universal piece is related to the constant term in the entropy scaling for a block-Toeplitz correlation matrix with jump singularities in the symbol, and can only be extracted numerically.

It would be interesting to extend the studies to initial states with arbitrary fillings on both sides of the chain. The particular case of a half-filled ground state extending into the vacuum has been studied in detail for homogeneous hoppings [57, 58], featuring a half-chain entropy growth of $1/4 \ln t$, which can be derived by a combination of hydrodynamics and curved-space CFT arguments [59]. For a dimerized chain, the ground state on the left corresponds to a completely filled band, and it is easy to guess that the prefactor would change to 1/6, which is indeed what we find in a numerical check. However, it is not straightforward to get the universal part of the entropy, as the modes on the two sublattices have nontrivial correlations in the initial state, which must be incorporated into the CFT description. In any case, a full analytic description of the entropy profile would also require explicit results on the constant term in block-Toeplitz determinants [55], which seems difficult to obtain.

Another question to be further explored is the fine structure of the front edge, which has been an intensively studied topic [60–69]. In most of the cases, one finds a universal behaviour of the correlations on a characteristic scale $t^{1/3}$ around the front edge, described by the Airy kernel [70]. While for the dimerized chain the density/current profiles are, up to rescaling, identical to the case of homogeneous evolution on the ballistic scale, it is not immediately clear whether this holds true for the edge regime. A systematic analysis of the correlations beyond the stationary phase approximation presented in Appendix A requires some further efforts.

Finally, one could address the domain-wall melting problem for interacting fermions with dimerized hoppings. In the homogeneous case, the problem can be solved via GHD for particular values of the interaction [11]. One finds then an effective description in terms of an inhomogeneous Luttinger liquid, and a logarithmic entropy growth with a prefactor 1/6 [71]. However, the dimerization would break the integrability of the time-evolution operator, hence it is unclear whether there is any underlying hydrodynamic picture and if the logarithmic growth of the entropy persists.

## Acknowledgments

The author thanks F. Ares, P. Calabrese, J. Dubail and S. Scopa for fruitful discussions.

**Funding information** This research was funded in whole by the Austrian Science Fund (FWF) Grant-DOI: 10.55776/P35434 and 10.55776/PAT3563424. For the purpose of open access, the authors have applied a CC BY public copyright licence to any Author Accepted Manuscript version arising from this submission.

# A    Stationary phase calculation

In this appendix we provide the derivation of the correlation matrix (12) in the hydrodynamic limit, which follows the lines of Ref. [62]. Rewriting (7) in block notation we have

$$\mathbf{C}_{m,n}(t) = \sum_{j \le 0} \mathbf{U}^{\dagger}_{m,j} \mathbf{U}_{j,n} \,. \tag{A.1}$$

Using the form of the propagator (11) and the identity

$$\sum_{j=1}^{\infty} e^{i(q-p)(2j-1)} = \frac{i}{2 \sin(q-p+i0)} \,, \tag{A.2}$$

we arrive at the expression

$$\mathbf{C}_{m,n}(t) = \int_{-\pi/2}^{\pi/2} \frac{dp}{\pi} \int_{-\pi/2}^{\pi/2} \frac{dq}{\pi} \frac{e^{iq(2n-1)} e^{-ip(2m-1)}}{-2i \sin(q-p+i0)} \mathbf{V}(p,q) \,, \tag{A.3}$$

where the matrix elements of the $2 \times 2$ matrix $\mathbf{V}(p,q)$ are given by

$$\begin{aligned}
V_{1,1} = V_{2,2}^* &= \cos(\omega_q t) \cos(\omega_p t) + \sin(\omega_q t) \sin(\omega_p t) e^{-i(\varphi_q - \varphi_p)} \,, \\
V_{1,2} = -V_{2,1}^* &= i \sin(\omega_q t) \cos(\omega_p t) e^{i\varphi_q} - i \cos(\omega_q t) \sin(\omega_p t) e^{i\varphi_p} \,.
\end{aligned} \tag{A.4}$$

The hydrodynamic limit corresponds to $m, n, t \gg 1$, with $m/t$ and $n/t$ kept fixed such that $|m-n|/t \ll 1$. In turn, this amounts to a stationary phase analysis of the double integral in (A.3), where the dominant contribution is due to the pole at $p = q$. Introducing the variables $Q = q - p$ and $P = (q+p)/2$, one can thus expand the oscillatory phases around $Q = 0$ via the group velocity $v_P = -\omega'_P$ as

$$\omega_p \approx \omega_P + \frac{Q}{2} v_P \,, \qquad \omega_q \approx \omega_P - \frac{Q}{2} v_P \,. \tag{A.5}$$

Furthermore, the products of trigonometric functions in (A.4) can be rewritten via sums with arguments $(\omega_p \pm \omega_q)t$. It is then easy to see, that the phase $(\omega_p + \omega_q)t \approx \omega_P t$ cannot be made stationary, whereas the one with the difference can be evaluated via the identity

$$\Theta(x) = -\int_{-\infty}^{\infty} \frac{dQ}{2\pi i} \frac{e^{-iQx}}{Q+i0} \,, \tag{A.6}$$

where $\Theta(x)$ is the Heaviside step function. Indeed, expanding the integrand in (A.3) around $Q = 0$, and extending the range of the $Q$-integral to infinity, one obtains to leading order

$$\mathbf{C}_{m,n}(t) = \int_{-\pi/2}^{\pi/2} \frac{dP}{\pi} e^{iP(2n-2m)} \mathbf{W}(P) \,, \tag{A.7}$$

with matrix elements

$$\begin{aligned}
W_{1,1} = W_{2,2} &= \frac{1}{2} [\Theta(v_P t - x) + \Theta(-v_P t - x)] \,, \\
W_{1,2} = W_{2,1}^* &= \frac{e^{i\varphi_P}}{2} [\Theta(v_P t - x) - \Theta(-v_P t - x)] \,,
\end{aligned} \tag{A.8}$$

where $x = m + n - 1$. This is precisely the result reported in (12).

One should stress that, in passing from (A.4) to (A.8), we substituted $\varphi_q \approx \varphi_p \approx \varphi_P$, i.e. we assumed that the phase varies slowly and factors of $\varphi'_P$ are negligible against the factors $v_P t$ and $x$. This is indeed true for larger values of $\delta$, however, for $\delta \ll 1$ the phase changes very quickly around the edges of the reduced Brillouin zone $P \to \pm \pi/2$, as shown in Fig. 9. In particular, in the limit $\delta \to 0$ one has $\varphi'_P \approx (\pi/2 - P)/\delta$ around the right edge, which shows that there is no smooth transition towards the case $\delta = 0$, and it is the origin of the strong oscillations observed in the entropy profiles for small dimerizations.

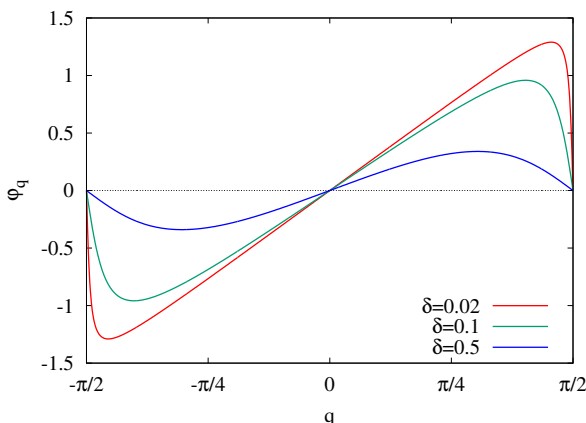

Figure 9: Variation of the phase $\varphi_q$ within the reduced Brillouin zone for various $\delta$.

## B Block Toeplitz determinants

In this appendix we summarize the results of Ref. [55] for the asymptotics of block Toeplitz determinants with piecewise continuous symbols. The result is then applied to find the leading order term in (33), using the determinant representation of the entropy [50]

$$S = \frac{1}{2\pi i} \oint_\Gamma d\lambda\, s(\lambda) \frac{d}{d\lambda} \ln \det(\lambda \mathbb{1}_L - C_L),\tag{B.1}$$

where $s(\lambda) = -\lambda \ln \lambda - (1-\lambda)\ln(1-\lambda)$ is the entropy density and the contour encircles the eigenvalues of the reduced correlation matrix $C_L$ contained in the interval $[0,1]$. We will assume $T_L = \lambda \mathbb{1}_L - C_L$ to be a block Toeplitz matrix, composed of matrix elements

$$T_{m,n} = \int_{-\pi}^{\pi} \frac{dk}{2\pi}\, \phi(k)\, e^{i(n-m)k},\tag{B.2}$$

with $1 \leq m,n \leq L$, and a piecewise continuous matrix symbol $\phi(k)$ of size $M \times M$, with discontinuities at momenta $k_s$ and $s = 1,\dots,R$.

Let us assume that the symbol can be factorized as

$$\phi(k) = \phi_0(k) \prod_{s=1}^{R} \phi_s(k), \qquad \phi_s(k) = e^{iB_s(k-k_s-\pi)},\tag{B.3}$$

where $\phi_0(k)$ is a smooth invertible function, and for $\phi_s(k)$ the $2\pi$-periodic momenta are chosen from the interval $k_s < k < k_s + 2\pi$, such that it describes a jump around $k_s$. In fact, it can be shown that, up to a similarity transformation, one has [55]

$$B_s \simeq \frac{1}{2\pi i} \ln[\phi^{-1}(k_s+0)\phi(k_s-0)],\tag{B.4}$$

where $\phi(k_s \pm 0)$ is the symbol on the right/left hand side of the jump. The asymptotics of the block Toeplitz determinant then reads [55]

$$\det T_L = F^L L^\Omega E_1 E_2,\tag{B.5}$$

where the first term corresponds to the Szegö-Widom limit theorem with [72,73]

$$F = \exp\left(\int_{-\pi}^{\pi} \frac{dk}{2\pi} \ln \det \phi_0(k)\right).\tag{B.6}$$

Note that $F$ is responsible for the extensive term in the entropy, which must vanish in our case. The exponent of the relevant power-law term is given by

$$\Omega = -\sum_{s=1}^{R}\sum_{j=1}^{M}\beta_{s,j}^2\,, \tag{B.7}$$

where $\beta_{s,j}$ are the eigenvalues of the matrices in (B.4). The constant terms are given by

$$E_1 = \prod_{s=1}^{R}\prod_{j=1}^{M}G(1+\beta_{s,j})G(1-\beta_{s,j})\,, \tag{B.8}$$

$$E_2 = \det\left(T(\phi)T^{-1}(\phi_R)\dots T^{-1}(\phi_1)T^{-1}(\phi_1^{-1})\dots T^{-1}(\phi_R^{-1})T(\phi^{-1})\right)\,, \tag{B.9}$$

where $G$ is the Barnes $G$-function, and $T(\phi_k)$ is a Toeplitz operator acting on a semi-infinite domain, with corresponding block matrix symbol $\phi_k$.

In turn, the constant $E_1$ has precisely the same form as for a scalar symbol [74], with the eigenvalues at the various jumps contributing independently. In sharp contrast, the constant $E_2$ couples the various jumps and has the form of an operator determinant. While for scalar symbols its general form is known explicitly [74], evaluating it in the block-Toeplitz case is highly non-trivial even for smooth symbols, and there are only few examples that have been treated via Riemann-Hilbert analysis [75,76]. We thus restrict ourselves to the calculation of the power-law term $\Omega$ in (B.7), which depends only on the eigenvalues of the matrices in (B.4).

The symbol of the block-Toeplitz matrix associated to the correlation matrix (12) has jumps at $\pm k_{\pm}$ (with $k_{\pm} = 2q_{\pm}$ being the sublattice momentum), and the relevant limits of the symbol are obtained as

$$\phi(k_{\pm}\mp 0)=\frac{1}{2}\begin{pmatrix} 2\lambda-1 & -e^{i\varphi_{k_{\pm}}} \\ -e^{-i\varphi_{k_{\pm}}} & 2\lambda-1 \end{pmatrix}, \qquad \phi(-k_{\pm}\pm 0)=\frac{1}{2}\begin{pmatrix} 2\lambda-1 & e^{i\varphi_{k_{\pm}}} \\ e^{-i\varphi_{k_{\pm}}} & 2\lambda-1 \end{pmatrix}, \tag{B.10}$$

whereas on the other side of the respective jumps the symbol is trivial, $\phi(k_{\pm}\pm 0) = \phi(-k_{\pm}\mp 0) = \lambda\mathbf{1}$. Diagonalizing the corresponding matrices (B.4) this yields

$$\beta_{s,1}^2 = -\frac{1}{4\pi^2}\ln^2\left(\frac{\lambda-1}{\lambda}\right), \qquad \beta_{s,2}^2 = 0\,, \tag{B.11}$$

such that for each $s$ one of the eigenvalues vanishes, while the other has precisely the same form as in the calculation for the ground state [50]. Since the number of jumps is now doubled, so is the prefactor of the logarithm, which yields (33). Note that this remains true even in the limit $k_- \to 0$, where two Fermi points merge. One has then only three jumps, where the values of $\beta_{s,j}^2$ for $s = 1,3$ are unchanged. Instead, for $s = 2$ one needs the eigenvalues of the matrix $\phi^{-1}(k_-+0)\phi(-k_--0)$, which yields

$$\beta_{2,1}^2 = \beta_{2,2}^2 = -\frac{1}{4\pi^2}\ln^2\left(\frac{\lambda-1}{\lambda}\right), \tag{B.12}$$

and thus the value of $\Omega$ remains unchanged.

## C  Particle number fluctuations for $\delta \to 1$

Here we compute the particle number fluctuation in an interval $A = [1, L]$, for a block-Toeplitz correlation matrix of the form (12), with the Fermi points $q_{\pm}$ assumed to be fixed. For a free-fermion chain, the second cumulant of the particle number can be evaluated as

$$\langle \delta N_A^2 \rangle = \operatorname{Tr} C_A(1-C_A)\,. \tag{C.1}$$

We will be interested in the limit $\delta \to 1$, such that the phase vanishes, $\varphi_q \to 0$, and the expression of the correlation matrix simplifies to

$$C_{m,n} = \begin{cases} \mathcal{S}(r), & r \text{ even,} \\ -i\,\mathcal{C}(r-1), & r \text{ odd, } m \text{ odd,} \\ -i\,\mathcal{C}(r+1), & r \text{ odd, } m \text{ even,} \end{cases} \tag{C.2}$$

where $r = n - m$ and we introduced

$$\mathcal{S}(r) = \frac{\sin(q_+ r) - \sin(q_- r)}{\pi r}, \qquad \mathcal{C}(r) = \frac{\cos(q_+ r) - \cos(q_- r)}{\pi r}. \tag{C.3}$$

Note that in the limit $r = 0$ one has $\mathcal{S}(0) = (q_+ - q_-)/\pi$ and $\mathcal{C}(0) = 0$.

Writing out the trace in (C.1) one has

$$\sum_{n \in A}(C_{n,n} - C_{n,n}^2) - 2 \sum_{m < n \in A} |C_{m,n}|^2, \tag{C.4}$$

and introducing $q_F = q_+ - q_-$ one has for the first sum

$$\sum_{n \in A}(C_{n,n} - C_{n,n}^2) = L\left[\frac{q_F}{\pi} - \left(\frac{q_F}{\pi}\right)^2\right]. \tag{C.5}$$

Furthermore, carrying out the second sum along the diagonals with $r$ fixed one obtains

$$\begin{aligned} \sum_{m < n \in A} |C_{m,n}|^2 &= \sum_{\substack{r=0 \\ r \text{ even}}}^{L-2}(L-r)\mathcal{S}^2(r) + \sum_{\substack{r=1 \\ r \text{ odd}}}^{L-1}\left[\frac{L-r+1}{2}\mathcal{C}^2(r-1) + \frac{L-r-1}{2}\mathcal{C}^2(r+1)\right] \\ &= \sum_{\substack{r=0 \\ r \text{ even}}}^{L-2}(L-r)[\mathcal{S}^2(r) + \mathcal{C}^2(r)], \end{aligned} \tag{C.6}$$

where we used the alternating structure of the matrix elements for odd $r$, and changed variables in the second sum. This way one is left with a sum over even $r$ only, and the summand can be simplified using

$$\mathcal{S}^2(r) + \mathcal{C}^2(r) = \frac{2 - 2\cos[(q_+ - q_-)r]}{\pi^2 r^2} = \frac{4\sin^2 \frac{q_F r}{2}}{\pi^2 r^2}. \tag{C.7}$$

Since this expression decays as $1/r^2$, after multiplying with $L - r$ the two pieces of the sum in (C.6) delivers a linear as well as a logarithmic contribution in $L$. The extensive term can be evaluated by exchanging $r = 2j$, extending the sum to infinity and using

$$\sum_{j=1}^{\infty} \frac{\sin^2(q_F j)}{\pi^2 j^2} = \frac{1}{2}\left[\frac{q_F}{\pi} - \left(\frac{q_F}{\pi}\right)^2\right]. \tag{C.8}$$

Hence, taking the factor $-2$ in (C.4) into account, the extensive part exactly cancels with the diagonal sum in (C.5). Note, however, that if we are interested in the $\mathcal{O}(1)$ contributions to the fluctuations, one should keep also the $1/L$ corrections, which originate from extending the sum to infinity. Indeed, one has

$$\sum_{j=L/2}^{\infty} \frac{\sin^2(q_F r)}{r^2} \approx \sum_{j=L/2}^{\infty} \frac{1}{2r^2} = \frac{\psi^{(1)}(L/2)}{2} \approx \frac{1}{L}, \tag{C.9}$$

where in the first step we replaced the heavily oscillating sine function by $1/2$, and we used the asymptotics of the polygamma function $\psi^{(1)}$.

Putting everything together, the fluctuations read

$$\langle \delta N_A^2 \rangle = \frac{2}{\pi^2} + \frac{4}{\pi^2} \sum_{j=1}^{L/2-1} \frac{\sin^2(q_F j)}{j} . \tag{C.10}$$

Note that this is precisely twice the result for an interval of size $L/2$ in a homogeneous ground state at filling $q_F$ [40, 56]. The sum can be further evaluated as

$$\sum_{j=1}^{L/2-1} \frac{\sin^2(q_F j)}{j} \approx \frac{1}{2}(\psi(L/2) + \ln(2\sin q_F) + \gamma), \tag{C.11}$$

where $\psi$ is the digamma function. In turn, up to $1/L$ corrections, one finds the result

$$\langle \delta N_A^2 \rangle = \frac{2}{\pi^2}[\ln(L\sin q_F) + \gamma + 1]. \tag{C.12}$$

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
