# Peer review of "Domain-wall melting and entanglement in free-fermion chains with a band structure"

_SciPost Physics Core, doi:SciPost Phys. Core 8, 069 (2025)_

## Round 1 · Referee Report · Anonymous (Referee 1) · 2025-8-18

Strengths

  1. Clear and correct analysis of domain-wall melting in free-fermion chains with periodic hopping.

  2. Analytical results for density, current, and entanglement are well supported by numerics.

  3. The universal vs non-universal contributions to the entanglement entropy are carefully distinguished.

  4. The study is physically motivated, with a proper understanding of the multi-band effects.

Weaknesses

  1. The work is basically incremental, and does not introduce fundamentally new concepts.

  2. Some aspects of the study only rely on numerical simulations.

  3. The multi-band discussion is very brief and could be expanded; there are no studies of density or current, and no hydrodynamic treatment is provided.

  4. Edge behavior and finer features of the front are only qualitatively mentioned.

Report

The manuscript presents a careful study of domain-wall melting in free-fermion chains with periodic hopping amplitudes, focusing on both density/current profiles and entanglement dynamics.

The author shows that in dimerized chains, the hydrodynamic evolution of particle density and current can be obtained from the homogeneous case via a simple velocity rescaling. For entanglement entropy, he identifies a doubling of the universal logarithmic growth due to two filled quasiparticle bands and quantifies non-universal contributions via block-Toeplitz matrices. The approach is supported by extensive numerical checks.

The work is for sure relevant to the community studying quantum quenches, integrable systems, and entanglement dynamics. The clarity of the exposition makes the results accessible and reproducible.

However, overall the work is incremental, although it provides a clear and correct picture of the physics, with well-explained methods and results.

Minor improvements could include a more detailed discussion of the multi-band case, especially with respect to a possible generalisation fo the hydrodynamic approach.

Recommendation

Publish (easily meets expectations and criteria for this Journal; among top 50%)

  • validity: top
  • significance: high
  • originality: high
  • clarity: top
  • formatting: excellent
  • grammar: excellent

Author:  Viktor Eisler  on 2025-09-18  [id 5832]

(in reply to Report 1 on 2025-08-18)

Minor improvements could include a more detailed discussion of the multi-band case, especially with respect to a possible generalisation fo the hydrodynamic approach.

We thank the referee for his/her comments towards improving our manuscript. Following a similar suggestion from the other referee, we extended the discussion of the three-band case. In the revised version, we show the particle density along with the entropy profile in Fig. 8. The density shows similar features observed in the entropy, namely there is a second front edge appearing around $|\nu|=v_2$. Note also that there is a fine structure emerging, i.e. the density is not homogeneous within a cell. This indicates that the hydrodynamic limit of the correlations must have a more complicated structure than the one in (12) for the dimerized case. In particular, the diagonal terms of the respective 3x3 matrix must be unequal. These features are now also discussed in the text.

The above considerations already show, that the generalization of the hydrodynamic approach is more involved. Although the analogue of Eq. (12) could in principle be worked out, the analytical expressions for the density and current are likely to be very cumbersome. In particular, there seems to be no direct relation to the domain-wall melting problem in the homogeneous chain. Hence, it is not a priori clear, whether one could write down a corresponding inhomogeneous Dirac theory for the calculation of the entropy, especially because of the fine-structure observed in the density. These issues are clearly interesting and require further investigations.

---

## Round 1 · Referee Report · Anonymous (Referee 2) · 2025-8-26

Report

The manuscript investigates transport in one-dimensional free-fermionic chains initialized in a half-filled domain wall state. Compared to earlier results in the literature, the novel feature here is that the fermions have a gapped band structure. The authors show that the prefactor in the entropy evolution changes and appears to be proportional to the number of bands. This conclusion is supported by numerical simulations, hydrodynamic arguments, and the asymptotic analysis of block-Toeplitz determinants.

The results are interesting, and the manuscript is well written. I have only a few minor suggestions that, if addressed, could further improve the fullness of the presentation:

  1. In Eqs. (26)–(28), the authors provide a review of inhomogeneous CFT. It would be very helpful to see how the current (22) can be derived within this framework, especially since it seems that for particle transport only the top band contributes, unlike in the case of entanglement spreading.

  2. Along similar lines, could the authors please provide the particle density profile corresponding to the data in Figure 8? Would one observe the same step-like features there?

  3. For clarification: when the authors state that $S_0(\zeta)$ is evaluated numerically, is this done using Eqs. (57) and (58)? If so, it seems possible to avoid fitting the parameters $\alpha$, $\beta$, and $\phi$ from Eq. (36), and instead evaluate them directly from Eqs. (57) and (58).

  4. Finally, would it be possible to provide a hydrodynamic derivation of the particle number fluctuations?

Overall, I believe the manuscript makes a valuable contribution. After these minor corrections, I would recommend publication.

Recommendation

Ask for minor revision

  • validity: top
  • significance: ok
  • originality: ok
  • clarity: top
  • formatting: excellent
  • grammar: excellent

Author:  Viktor Eisler  on 2025-09-18  [id 5831]

(in reply to Report 2 on 2025-08-26)

We thank the referee for the careful reading of the manuscript and for the insightful comments. Below we address the referee's questions and provide some clarifications, indicating the changes to the manuscript.

In Eqs. (26)–(28), the authors provide a review of inhomogeneous CFT. It would be very helpful to see how the current (22) can be derived within this framework, especially since it seems that for particle transport only the top band contributes, unlike in the case of entanglement spreading.

It is not immediate to see the expression of the current from the theory in Eq. (26)-(28), as it is formulated in imaginary time. However, to get expectation values of local observables, such as the density and current, one does not need to write down this effective theory, as they already follow from the form (12) of the correlation matrix. The inhomogeneous Dirac theory is required only for the derivation of the entanglement entropy, which is a nonlocal object, and thus depends on the global structure of the underlying metric.

One actually has contributions from both bands to all of the discussed observables. This might not be so apparent in the form (22) and (24) due to symmetry reasons. However, looking at the form (12) of the correlation matrix, it is immediately clear that the positive momenta correspond to quasiparticles from the lower band, while the negative ones to those from the upper band, as stressed below Eq. (13).

Along similar lines, could the authors please provide the particle density profile corresponding to the data in Figure 8? Would one observe the same step-like features there?

We thank the referee for this suggestion, in the revised version we show the particle density along with the entropy profile in Fig. 8. The density shows indeed similar features, namely there is a second front edge appearing around $|\nu|=v_2$. Note also that there is a fine structure emerging, i.e. the density is not homogeneous within a cell. This indicates that the hydrodynamic limit of the correlations must have a more complicated structure than the one in (12) for the dimerized case. In particular, the diagonal terms of the respective 3x3 matrix must be unequal. These features are now also discussed in the text.

For clarification: when the authors state that $S_0(\zeta)$ is evaluated numerically, is this done using Eqs. (57) and (58)? If so, it seems possible to avoid fitting the parameters $\alpha$, $\beta$, and $\phi$ from Eq. (36), and instead evaluate them directly from Eqs. (57) and (58).

Unfortunately, although Eqs. (57)-(58) is a very nice result for block-Toeplitz matrices from a mathematical point of view, they do not allow for a numerical evaluation of the constant term. Indeed, while (57) has the same form as the corresponding term in case of a scalar symbol, (58) is a determinant expression that involves Toeplitz operators on a semi-infinite domain. We are not aware of any results, where this expression could be evaluated in closed form for a piecewise continuous symbol. Instead, we evaluate $S_0(\zeta)$ as described in the text around Eq. (33), i.e. assuming a correlation matrix of the form (12) with constant Fermi points $q_\pm(\zeta)$, inserting this into (9) to get the entropy, and subtracting the log contribution.

Finally, would it be possible to provide a hydrodynamic derivation of the particle number fluctuations?

The universal contribution to the fluctuations can indeed be evaluated using the same machinery as for the entanglement entropy. After bosonizing the theory, one ends up with an inhomogeneous Luttinger liquid, and the full counting statistics then follows as a two-point function of vertex operators, see e.g. Ref. [71]. Unfortunately, the bottleneck is again to obtain the non-universal contribution. We were able to calculate this in the limit of $\delta \to 1$ (see Appendix C), but failed to get an analytical result for general $\delta$.

---

## Round 2 · Author Response

Dear Editor,
we are resubmitting the manuscript after minor revision, and addressing all the issues raised by the referees.
Best regards,
Viktor Eisler
we are resubmitting the manuscript after minor revision, and addressing all the issues raised by the referees.
Best regards,
Viktor Eisler

---

## Round 2 · List of Changes

- extended discussion of the three-band case
- density profile added in Fig. 8, and corresponding description in the text

---

## Editorial Decision

published